# In Vitro and In Vivo Evaluation of Nanoemulsion Containing Vegetable Extracts

**Pedro Alves Rocha-Filho [1,\*]** (ID)**, Marcio Ferrari [2], Monica Maruno [3], Odila Souza [1] and Viviane Gumiero [1]**

[1] Department of Pharmaceutical Sciences, Faculty of Pharmaceutical Sciences of Ribeirão Preto, University of São Paulo, Avenida do Café, s/n, Bairro Monte Alegre, Ribeirão Preto, SP 14040-903, Brazil; odilinha@yahoo.com.br (O.S.); vcgumiero@hotmail.com (V.G.)

[2] College of Pharmacy, Federal University of Rio Grande do Norte, Rua Gustavo Cordeiro de Farias, s/n, Petrópolis, Natal, RN 59012-570, Brazil; ferrarimarcio@uol.com.br

[3] Pharmacy Course Coordination, Centro Universitário Barão de Mauá, R. Ramos de Azevedo, 423, Jardim Paulista, Ribeirão Preto, SP 14090-180, Brazil; monica.maruno@baraodemaua.br

\* Correspondence: pedranjo@fcfrp.usp.br; Tel.: +55-16-3315-4214

**Abstract:** Oil/Water nanoemulsions were obtained, employing PEG castor oil derivatives/fatty esters surfactant, babassu oil, and purified water from a study based on phase diagrams. The nanoemulsions had been prepared by a low energy process inversion phase emulsion. Different parameters, such as order of addition of the components, temperature, stirring speed, and time, were studied to prepare O/W nanoemulsions. The influence of vegetable extract addition on size distribution of nanoemulsions was also analyzed. Evaluation of the nanoemulsions was studied in vitro by HET-CAM and RDB methods. Stable transparent bluish O/W babassu oil nanoemulsion were obtained with surfactant pair fatty ester/PEG-54 castor oil, in an $HLB_{required}$ value = 10.0 and with a particle droplet size of $46 \pm 13$ nm. Vegetable extract addition had not influenced nanoemulsion's stability. The results obtained for in vitro and in vivo nanoemulsion evaluation, based on the hydration and oiliness, and pH of the skin, shows O/W nanoemulsions as potential vehicle for topical application.

**Keywords:** nanoemulsions; babassu oil; vegetable extract; efficacy evaluation

## 1. Introduction

For cosmetics products, nanoemulsions are preferable and more stable than macroemulsions, have good spreadability, and facilitate penetration of actives into the skin. The interest in nanoscale emulsion has been growing considerably in recent decades, due to its specific attributes, such as high stability, attractive appearance and drug delivery properties; therefore, performance is expected to improve using a lipid-based nanocarrier.

Nanoemulsions have recently become increasingly important as potential vehicles for the controlled delivery of cosmetics, and for the optimized dispersion of active ingredients, in particular, skin layers. Due to their lipophilic interior, nanoemulsions are more suitable for the transport of lipophilic compounds than liposomes. Similar to liposomes, they support the skin penetration of active ingredients, and thus, increase their concentration in the skin. Another advantage is the small-sized droplet, with its high surface area, allowing effective transport of the active to the skin. Furthermore, nanoemulsions gain increasing interest due to their own bioactive effects. This may reduce the transepidermal water loss (TEWL), indicating that the barrier function of the skin is strengthened. Nanoemulsions are acceptable in cosmetics, because there is no inherent creaming, sedimentation,

flocculation, or coalescence that is observed with macroemulsions. Nanoemulsions are generated by different approaches: the so-called high-energy and low-energy methods.

Pereira et al. [1] used the oils of *Rubus idaeus* (raspberry seed oil), *Passiflora edulis* (seed oil), and *Prunus persica* (peach kernel oil), associated with lanolin derivatives, for Oil/Water nanoemulsions obtention. The authors verified that these lanolin derivatives cause alterations in the particle size, but they can be used for topical application.

Most recently Rocha-Filho et al. [2] had studied the action of tea tree oil and lavender oil on the stability of rice bran oil nanoemulsions. The authors observed that the presence of lavender essential oil causing a decrease in particle size, so, promoting the stabilization of the dispersed system.

The purpose of this article was to produce nanoemulsions, containing babassu oil and different vegetables extracts, through a low-energy method. So, babassu oil (*Orbignya oleifera*) was the oil phase, and is used in the treatment of various skin disorders, due to its anti-inflammatory, antiseptic, and healing properties. The aqueous phase was composed for *Areca catechu* seed extract, *Glycyrrhiza glabra* root extract (licorice), *Portulaca oleracea* (portulaca) extract and purified water. These extracts were recommended for skin recovery, mainly because of their antioxidant, anti-inflammatory, antibacterial, astringent, anti-hyaluronidase, melanogenesis inhibitor, anti-irritant, and healing promoter activities.

This study had several objectives: (1) to develop nanoemulsions containing babassu oil; (2) to incorporate vegetable extracts in the formulations, such as *Areca catechu* seed extract (Ar), *Glycyrrhiza glabra* (Al) and *Portulaca oleracea* extract; (3) to perform in vitro nanoemulsion evaluation by HET-CAM and RDC methods; and (4) to evaluate nanoemulsions' effect on skin hydration, oiliness, and pH, in vivo.

## 2. Materials

Oil phase: *Orbignya oleifera* seed oil (Crodamazon® Babaçu CO-Croda Brasil); surfactants: sorbitan monooleate (Span® 80) (HLB value = 4.3); PEG-54 castor oil (Ultramona® R540) (Hydrophilic Lipophilic Balance value = 14.4) (all surfactants Oxiteno Brasil); vegetable hydrophilic extracts: *Areca catechu* seed extract; *Glycyrrhiza glabra* (licorice) root extract and *Portulaca oleracea* extract (employed as provided by Lipo do Brasil); and aqueous phase: purified water.

## 3. Methods

### 3.1. Formulation Studies

#### 3.1.1. Nanoemulsion Development

The emulsions were prepared by the emulsion phase inversion method (EPI). The aqueous and oil phases were heated separately at $75 \pm 2$ °C. Then, the aqueous phase was added slowly over to the oily phase containing surfactants, and constant stirring (600 rpm) (Mod-Fisaton mechanical stirrer-713, Fisaton, São Paulo, SP, Brasil) until mixture reaches room temperature ($25 \pm 2$ °C) [3].

#### 3.1.2. Vegetable Extract Addition on Babassu Oil Nanoemulsions

Nanoemulsions were prepared by the cited method (3.1.1) and vegetable extracts were added separately at the recommended concentration used in scientific literature, 3.0%.

#### 3.1.3. Droplet Size and Polydispersity Index Determination

Droplet size diameter and polydispersity index of the nanoemulsions were determined by dynamic light scattering (DLS) (Nanosizer Malvern ZS, Worcestershire, UK) at a scattering angle of 173°, and the samples were diluted in purified water in the proportion 1:100, at 25 °C.

### 3.1.4. pH Evaluation

Nanoemulsion (1.0 g) was homogenized with 9.0 g of purified water, and the pH value was measured by inserting the electrode (pH meter Analion-Mod. PM608, Analion, Ribeirão Preto, SP, Brazil) directly into the sample solution at $24.0 \pm 2.0\,^{\circ}$C [4].

### 3.1.5. Electrical Conductivity Evaluation

The electrical conductivity of the emulsions was evaluated with the conductivity Digimed (DM32, Digimed, São Paulo, Brazil) calibrated with standard solution: the electrode was inserted directly on the sample at $24.0 \pm 2.0\,^{\circ}$C [4,5].

### 3.1.6. Refractive Index

The RI of the system was measured (in triplicates) by an Abbe refractometer (Bausch and Lomb Optical Company, Rochester, NY, USA) by placing one drop of the formulation on the slide, at $24 \pm 2\,^{\circ}$C. The refractometer was calibrated with purified water (1.333) [6].

### 3.1.7. Freeze-Defrost Cycles

The nanoemulsions were subjected to a temperature of $45 \pm 5\,^{\circ}$C for 24 h, and after, to a temperature of $4 \pm 2\,^{\circ}$C also for 24 h, thereby completing a cycle. Macroscopic evaluation was made 24 h after nanoemulsions' preparation, and at the end of the 6th cycle (day 12) [7].

## 3.2. In Vitro Nanoemulsion Evaluation

### 3.2.1. Hen's Egg Test on the Chorioallantoic Membrane (HET-CAM)

This method corresponds to a modification of the method described by [8]: embryonated chicken eggs were purchased from commercial hatcheries 10 days after fertilization and were maintained at $37 \pm 2\,^{\circ}$C. With scissors, a small incision was made in the center of the upper part of the shell of the egg. Subsequently, the whitish membrane that adhered to the inside of the egg was pulled out, to make clear the chorioallantoic membrane transparent with blood vessels.

The samples (0.3 g) were applied on the egg chorioallantoic membrane, and after 20 s of contact, it was washed with saline solution (5 mL). It was determined, the time (seconds) at which signs of irritation appear. Sodium lauryl sulfate (SLS) (10.0% solution) was used as a positive control, and saline solution (0.9%) and purified water, both, as a negative control.

The irritations produced were evaluated according to Luepke scale (Table 1), and were classified according Table 2. The test for each sample was performed in quadruplicate.

**Table 1.** Luepke scale for the appearance of the phenomena as function of time.

| Phenomena | Time (T) | | |
|---|---|---|---|
| | T ≤ 30 s | 30 s > T ≤ 2 min | 2 min > T ≤ 5 min |
| Hyperemia | 5 | 3 | 1 |
| Hemorrhage | 7 | 5 | 3 |
| Coagulation | 9 | 7 | 5 |

**Table 2.** Classification of products according to the scores of the phenomena.

| HET-CAM Index | Classification |
|---|---|
| N ≤ 1 | practically not irritant |
| 1 < N ≥ 5 | slightly irritant |
| 5 < N ≥ 9 | moderately irritant |
| N > 9 | irritant |

### 3.2.2. Irritation Test in Red Blood Cell (RBC) System Cellular Model

The human venous blood samples were freshly collected and put into a test tube containing anticoagulant (EDTA-Na$_2$ 10.0%) (CEP/FCFRP n$^r$ 204/2011). For the calculation of H$_{50}$ (effective concentration that causes 50.0% of hemolysis), formulations and surfactant solution separately were diluted (triplicates) to 0.02, 0.05, 0.1, 0.2, 0.4, 0.5, 0.6, 0.7, and 0.8 g/mL, in phosphate buffered saline at 10.0%, then, added 50.0 µL of blood, and the tubes were homogenized and incubated for 90 min at room temperature [9]. The samples were centrifuged at 2000 rpm for 5 min, and the supernatant was removed to measure the absorbance (540 nm) against the blank (100.0% buffered solution containing red blood cells). The results were compared with a tube in which the cells were completely lysed by distilled water (positive control). The hemolytic activity of each sample was calculated by the formula:

$$H(\%) = \frac{Abs_{sample}}{Abs_{control}^{+}}$$

For hemolytic activity, samples with values higher than H$_{50}$ were analyzed based on red blood cells denaturation. The absorbance readings of the supernatants are performed at wavelengths 540 and 575 nm against the blank (test substance diluted in buffer). The value of the extinction measured at 575 nm ($\alpha$) is divided by the value of extinction measured at 540 nm ($\beta$) to obtain the ratio $\alpha/\beta$. This ratio is used to characterize the denaturing index (DI) of hemoglobin.

$$DI(\%) = (R_1 - R_i)/(R_1 - R_2) \times 100$$

where:

R$_1$—ratio $\alpha/\beta$ hemoglobin;
R$_i$—ratio $\alpha/\beta$ of the test substance;
R$_2$—ratio $\alpha/\beta$ 0.1% of the SLS.

The relationship between the concentration that causes 50.0% hemolysis (H$_{50}$) and the denaturing index (DI) is defined as the ratio of H$_{50}$/ID, and was calculated for each test substance; the irritation potential is classified according to Table 3.

**Table 3.** Red blood cell (RBC) system cellular model applied to determine potential irritation of nanoemulsions [9].

| Ratio (H$_{50}$/ID) | Classification |
|---|---|
| >100.0 | No irritant |
| ≥10.0 | Slightly irritant |
| ≥1.0 | Moderately irritant |
| ≥0.1 | Severe irritant |
| <0.1 | Maximum irritant |

### 3.3. In Vivo Nanoemulsion Evaluation

### 3.3.1. Anti-Inflammatory Activity Evaluation by Edema Ear Rats Induced by Croton Oil

The mice used were kind of "Swiss" females, weighing between 25–30 g. The animals were kept in light/dark cycles of 12 h, with access to water and feed during the experiment. The number of animals used in the experiment was 38, divided into 5 groups of 6 to 8 animals each. The project was approved by the Ethics Committee on Animals (Protocol n$^r$ 10.1.1180.53.5-FCFRP-USP, 02/04/2011).

The anti-inflammatory activity was tested by acute ear edema, after topical application of 20 µL of croton oil solution (5.0% *v/v*) in acetone on the inner surface of the right ear (the left ear was correspondent control to right ear), with the exception of group I (negative control), which received no

treatment. Thirty minutes after croton oil application (20 µL), groups II, III, IV and V were treated with only the vehicle, babassu oil nanoemulsions without vegetable extracts, babassu oil nanoemulsions with vegetable extracts, and dexamethasone solution (positive control—4.0 mg/mL), respectively. In groups I, II, and V, acetone (20 µL) was applied in the left ear, while acetone (20 µL) was applied for groups III and IV over formulation's vehicle (20 µL) (this vehicle was composed of water, surfactants, microbial preservative, and BHT).

Four hours later, the mice were sacrificed and the ear thickness was measured (mm). Then, ear circles of 6 mm were cut out and weighed on an analytical balance, to assess the intensity of mass edema (mg). In both methods, thickness and weight values for the right ears were discounted from the opposite side ears in all groups, and converted to a percentage, relative to the negative control [10,11].

### 3.3.2. Skin Hydration, Oiliness, and pH Evaluation

After approval by the Ethics Committee on Human (Protocol n$^r$ 204-FCFRP-USP, 04/02/2011, 07/09/2011), babassu oil nanoemulsion dermal activity assessment was held in a room with controlled relative humidity (60 ± 3%) and temperature (22 ± 2 °C).

(a)　*Inclusion and exclusion criteria for selecting volunteers* [12].

It was selected 30 healthy volunteers, female and male, aged between 18 and 30 years, and free of skin care product use for at least 30 days. Volunteers with skin diseases or hypersensitivity to any component of the formulation were not accepted.

(b)　*Application of formulations*

Six areas were defined on each volunteer's forearm, totaling 12 areas. The demarcated areas were divided according to the following groups (triplicate):

(1)　babassu oil nanoemulsion without plant extracts;
(2)　babassu oil nanoemulsion with plant extracts;
(3)　commercial nanoemulsion Mist Moisturizing Milk Ekos$^®$Cupuaçu (Natura, São Paulo, Brazil);
(4)　control areas (without application of any formulation).

Babassu oil nanoemulsion (50.0 mg) was applied in a circular motion in a specific site of volunteer's forearm skin, and the assessment was made after 30, 60, 90, and 120 min. Skin hydration, pH, and oiliness measurements were performed in triplicate in each demarcated area with Corneometer$^®$ CM 820 equipment (Courage + Khazaka Electronic GmbH, Cologne, Germany) [13].

### 3.3.3. Skin Hydration Evaluation

The changes in capacitance were detected by a probe, and converted to hydration units ranging 0–150 arbitrary units (AU), where 0 corresponds to very dry skin and 150 to the very hydrated skin [14]. Skin hydration was calculated by the following equation:

$$HR\% = \frac{100 \times M_p}{M_c}$$

where:

HR% = relative hydration;
Mp = average capacitance readings of product application areas;
Mc = average capacitance of the readings of the control region.

### 3.3.4. Skin pH Value Assessment

The study was conducted using equipment pHmeter Skin$^®$ PH 900 (Courage + Khazaka) and the measures were carried in delimited areas of volunteer's forearm skin.

### 3.3.5. Skin Oiliness Evaluation

The equipment used for the determination of skin oils was Sebumeter®, and measures were made in volunteer's forearm skin in delimited areas. The measuring time is 30 s for each point measured. The transparency of the plastic strip was evaluated, thereby quantifying the presence of lipid content on the skin surface, and the result is expressed in g of fatty material/cm$^2$ [15].

### 3.4. Statistical Analysis

The results were presented as mean ± standard deviation (Microsoft Office Excel 2007 software). The analysis of values' variance was made, considering a 95% significance level, and non-parametric ANOVA (Prism Software GraphPad® Prism version 4.00), followed by multiple comparisons by Newman–Keuls test.

## 4. Results

### 4.1. Formulation Studies

Nonionic surfactants are widely selected for cosmetic formulations, because of lower potential skin irritation [16]. Surfactant pairs composed by PEG castor oil 54 Ethylene Oxyde/sorbitan monooleate at HLB$_{required}$ values of 10.0 produce the most stable emulsions for babassu oil, as it was present in the right corner of ternary diagram (Figure 1A).

From Figure 1B, it was observed that the derivatives' formulas are around point 36, numbered 37 to 44, and it can be observed that the nanoemulsions were translucent, clear, and with an intense bluish reflection, without macroscopic signs of instability (Figure 1C,D). On the other hand, if the amount of oil incorporated was increased, and surfactant quantity was reduced in the formula, it was observed as a loss in bluish color reflection, and loss of translucency of the nanoemulsions.

Formula 38 had a bluish reflection and translucent appearance, and after centrifuge test and thermal stress analysis, was macroscopically stable.

Areca, licorice and portulaca extracts were added separately at the recommended concentration in scientific literature (3.0%). Macroscopic analysis was performed, employing 24 h old nanoemulsions. Stable babassu oil nanoemulsions were obtained with 3.0% *w*/*w* vegetable extract added separately, or blended in and now designated as F-38J, with HLB value = 10.0 (sorbitan monooleate/PEG 54 castor oil added with three vegetable extracts), and were chosen for comparative studies. Figure 2 shows size distribution for both studied F-38 and F-38J nanoemulsions.

In order to determine physical and chemical differences between F-38 and F-38J samples, they were assessed for particle size distribution, polydispersity index, pH, electrical conductivity and refractive index methods, before and after thermic stress (Table 4).

There was no statistically significant difference ($p > 0.05$) between F-38 analyzed before and after the thermal stress test in all evaluated parameters. For F-38J nanoemulsions, it was observed that there is a statistically significant difference in polydispersity index and electrical conductivity values before and after the test.

Also, there was no statistically significant difference ($p > 0.05$) between the nanoemulsions F-38 analyzed before and after the freeze–defrost cycle test in the evaluated parameters (Table 4).

Babassu oil nanoemulsions, with and without extracts, were subjected to storage at temperatures of $4 \pm 2$ °C, $25 \pm 5$ °C and $45 \pm 5$ °C. The pH, electrical conductivity, refractive index values, particle size distribution, polydispersity index, and Ostwald ripening phenomenon were determined with samples that were 1, 7, 15, 30, 45, 60, 90, and 120 days old. For both F-38 and F-38F nanoemulsions stored at $4 \pm 2$ °C, $25 \pm 5$ °C, no statistically significant difference was observed from the first and after 120 days. Both formulas, when exposed at $45 \pm 5$ °C, showed a significant decrease in pH value after 120 days. For electrical conductivity, F-38J shows similar behavior as F-38: an increase in electrical conductivity values for samples stored at $45 \pm 5$ °C, while the samples stored at $4 \pm 2$ °C, and $25 \pm 5$ °C showed no statistically significant difference. These results indicate the presence of signs of instability

in the system with increasing temperature (45 ± 5 °C). These results corroborate to the results of [3] in the development of passion fruit and lavender oil nanoemulsions.

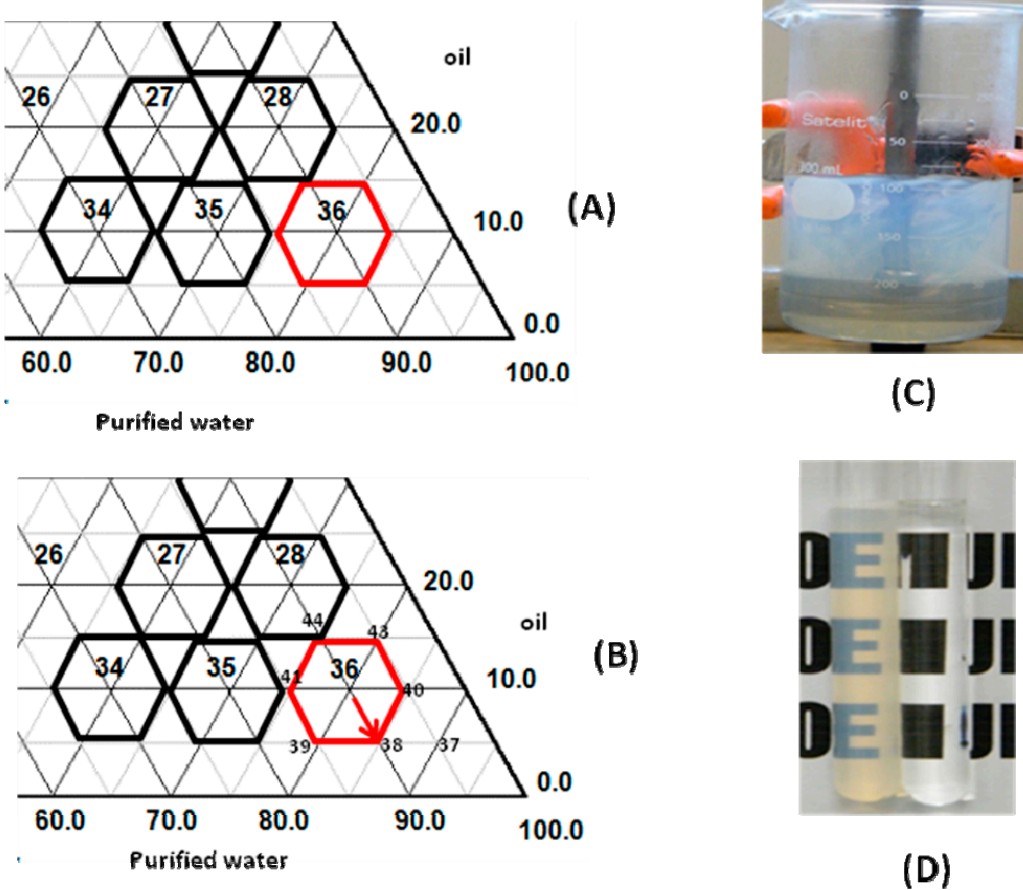

**Figure 1.** (**A**) Details of the phase diagram for babassu oil/water; (**B**) phase diagram—5.0% to 5.0%/surfactant (castor oil 54 EO/sorbitan monooleate, HLB value = 10.0); (**C**) nanoemulsion preparation; and (**D**) nanoemulsion.

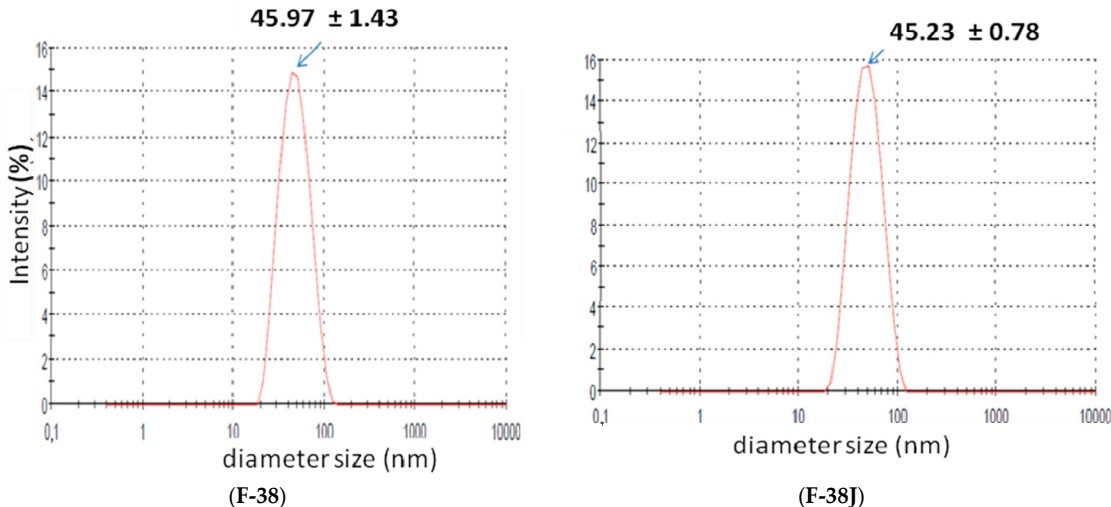

**Figure 2.** Particle size distribution comparison for babassu oil nanoemulsion without (**F-38**) and with (**F-38J**) vegetable extracts.

**Table 4.** Thermal stress and freeze–defrost analysis for F-38 and F-38J babassu nanoemulsions oil.

*Thermal Stress*

| Parameters | F-38 | | F-38J | |
|---|---|---|---|---|
| | **Before** | **After** | **Before** | **After** |
| droplets size (nm) | 45.97 ± 1.43 | 45.77 ± 0.65 | 45.23 ± 0.38 | 57.83 ± 6.15 |
| polydispersity index | 0.106 ± 0.004 | 0.093 ± 0.020 | 0.091 ± 0.008 | 0.137 ± 0.022 |
| pH value | 6.32 ± 0.07 | 6.27 ± 0.11 | 6.53 ± 0.03 | 6.23 ± 0.05 |
| electrical conductivity (μS/cm) | 283.25 ± 2.56 | 290.19 ± 3.77 | 406.12 ± 5.91 | 453.14 ± 6.7 |
| refractive index | 1.359 ± 0.001 | 1.361 ± 0.001 | 1.362 ± 0.001 | 1.371 ± 0.004 |

*Freeze–Defrost Cycles*

| Parameters | F-38 | | F-38J | |
|---|---|---|---|---|
| | **Before** | **After** | **Before** | **After** |
| droplets size (nm) | 45.97 ± 1.43 | 46.50 ± 0.60 | 45.23 ± 0.38 | 51.03 ± 477 |
| polydispersity index | 0.106 ± 0.004 | 0.103 ± 0.001 | 0.091 ± 0.008 | 0.125 ± 0.022 |
| pH value | 6.32 ± 0.07 | 6.42 ± 0.10 | 6.53 ± 0.03 | 6.45 ± 005 |
| electrical conductivity (μS/cm) | 283.25 ± 2.56 | 287.93 ± 3.98 | 406.12 ± 5.91 | 417.13 ± 11.19 |
| refractive index | 1.359 ± 0.001 | 1.359 ± 0.001 | 1.362 ± 0.001 | 1.363 ± 0.002 |

## 4.2. Nano Emulsion In Vitro Evaluation

### 4.2.1. Hen's Egg Test on the Chorioallantoic Membrane (HET-CAM)

The negative controls showed no change in the chorioallantoic membrane during the test. The positive control (SLS) was classified as severely irritant. It showed hyperemia and hemorrhage in under 30 s and 2 min, respectively, having a score of 10 on the scale (Table 5).

**Table 5.** Results for HET-CAM test.

| O/W and Test Solutions | Time | | | Score | Classification |
|---|---|---|---|---|---|
| | **Hyperemia** | **Hemorrhage** | **Coagulation** | | |
| F-38 | ≤30 s | ___ | ___ | 5 | slightly irritant |
| F-38J | ≤30 s | ___ | ___ | 5 | slightly irritant |
| Surfactants solution | ≤30 s | ___ | ___ | 5 | slightly irritant |
| Saline solution | ___ | ___ | ___ | 0 | practically not irritant |
| Purified water | ___ | ___ | ___ | 0 | practically not irritant |
| SLS solution | ≤30 s | 30 s < $T$ ≤ 2 min. | ___ | 10 | irritant |

In Table 5, we can see the occurrence of hyperemia for the nanoemulsions and also surfactants solution in a time less than 30 s, with no signs of hemorrhage and/or coagulation, corresponding to a score equal to 5, classifying the samples as slightly irritant. Then, both nanoemulsions F-38 and F-38J are safe, because the test did not show signs of hemorrhage and/or coagulation.

Our results are in accordance with Pereira [17] and Zanatta [18], in studies with nanoemulsions containing octyl methoxycinnamate, olive oil, grape seed oil, castor oil 30 EO, and sorbitan monooleate, which were considered slightly irritating, while the surfactants solution separately was moderately irritant. Zanatta [18] had employed buriti oil, sorbitan mono-oleate, and castor oil 40 EO.

Nanoemulsions containing surfactants derived from castor oil with different numbers of Polyoxyethylene (15, 30, 40 and 54 EO) and sorbitan monooleate were evaluated by the same method as Maruno [19], and the irritation potential was assessed as slightly irritant, reduced induced erythema, and demonstrated to be safe for cosmetics use.

These results indicate that in an emulsion, the surfactant is the most irritating component, and this characteristic increases with the number of ethylene oxides in the molecules.

### 4.2.2. Irritation Test in Red Blood Cell (RBC) System Cellular Model

This assay allows the quantification of adverse effects of isolated raw materials and finished products on the plasma membrane of red blood cells, and the consequent release of hemoglobin (hemolysis), quantified by denaturation of hemoglobin index, evaluating the hemoglobin oxidized form by spectrophotometry. The relationship between hemolysis and hemoglobin oxidation supplies an in vitro parameter, characterizing the effects of these substances [7,9].

In all nanoemulsion concentrations evaluated, lysis of red blood cells occurred, preventing the calculation of $H_{50}$, and consequently, the calculation of hemoglobin denaturation index, which enables the determination of the irritation degree of the emulsions. Then, no hemolytic activity was detected, and babassu oil nanoemulsions were classified as non-irritants, regardless of the addition or not of vegetable extracts.

Surfactant solution containing antimicrobial preservative and BHT presented hemolytic activity from 0.40 g/mL, indicating a dose-dependent activity. For the different tested concentrations, hemolytic activity was $0.99 \pm 0.12$ (0.40 g/mL), $6.03 \pm 0.50$ (0.50 g/mL), $12.09 \pm 0.52$ (0.60 g/mL), $52.76 \pm 0.55$ (0.70 g/mL), and $60.31 \pm 0.59\%$ (0.80 g/mL) (Figure 3).

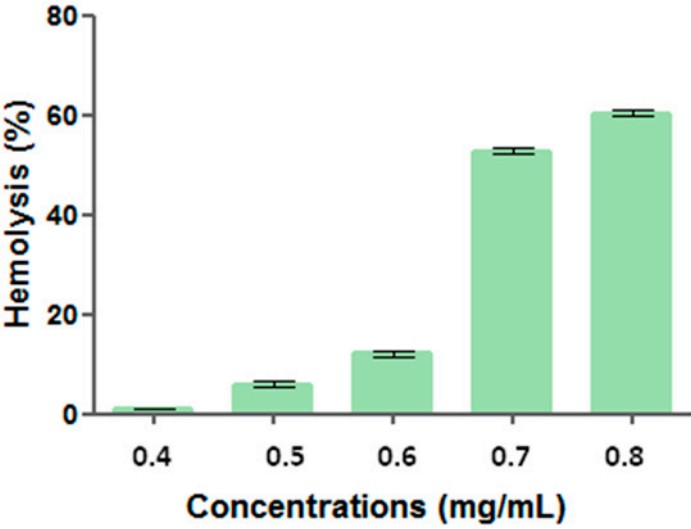

**Figure 3.** Hemolytic activity (%) versus surfactant solution concentration.

The formulations of babassu oil nanoemulsions evaluated by both HET-CAM and RBC method, were evaluated as slightly irritating, and non-irritating, respectively. Correlating the HET-CAM results, and those obtained by RBC, babassu oil nanoemulsions F-38 and F-38J may be indicated as safe for cosmetic use.

### 4.3. In Vivo Nanoemulsion Evaluation

### 4.3.1. Anti-Inflammatory Activity Evaluation by Edema Ear Rats Induced by Croton Oil

The ear edema induced by croton oil is a widely used model to assess the anti-inflammatory activity of steroidal and nonsteroidal drugs [10]. The main compound present in the croton oil (*Croton tiglium*) that acts as an inflammatory agent is 12-O-tetradecanoylphorbol-13-acetate (TPA). The application of the oil results in rapid accumulation of inflammatory cells, such as neutrophils and macrophages, to produce reactive oxygen species at the site [11].

After 4 hours of croton oil application in the ears of mice, a vivid hyperemia was visualized, and consequently, edema induction. For the ear thickness, the negative control already discounted from the left ear values presented $0.161 \pm 0.022$ mm of thickness (Figure 4). Groups II, III, IV, and V present a statistically significant difference ($p < 0.001$) compared to group I, with edema reduction of 25.97,

20.93, 44.67, and 66.93%. Groups II and III were the only groups that did not differ amongst the groups ($p > 0.05$).

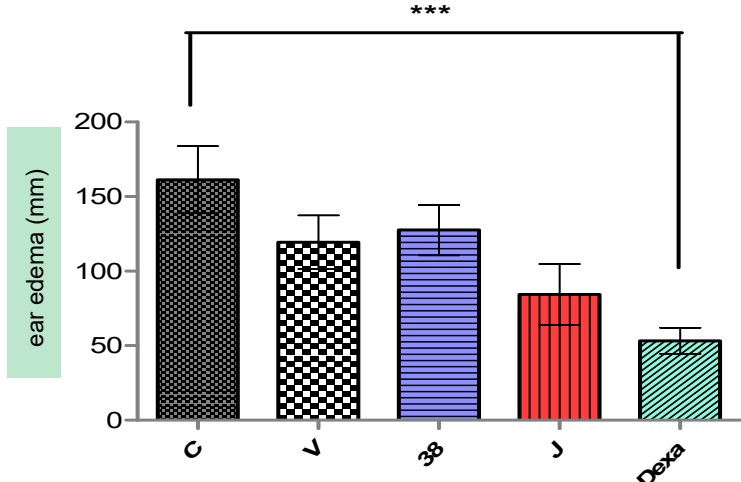

**Figure 4.** Ear edema induced by croton oil in untreated and treated animal groups. (**C**) Negative control; (**V**) vehicle; (**F-38**) nanoemulsion without extracts; (**F-38J**) nanoemulsion with extracts; (**Dexa**) dexamethasone solution. Values represent the mean and the respective standard deviations ($n = 6–8$). *** Significance level compared to the control group (**C**) ($p < 0.001$).

In relation to auricular mass, the negative control showed $5.9 \pm 0.63$ mg. Groups II, III, IV, and V have statistically significant differences compared to the negative control, with reduction of edema of 12.30, 16.09, 44.97, and 71.04%.

Thus, it is suggested that the chemical components in the F-38 and F-38J babassu oil nanoemulsions may be inhibiting the release of inflammatory mediators, or antagonizing the pharmacological receptors.

The results obtained by Sonneville-Aubrun et al. [20] in studies about the penetration rate of a nanoemulsion of 15.0% oil, in comparison to their corresponding macroemulsion, showed that the nanoemulsion penetrated significantly faster than the macroemulsion.

Nanoemulsions are an ideal system as vehicles for actives, even if they are insoluble in water. The small particle size allows for better adhesion of the active substance. In addition, spreadability, wetting, and penetration of the nanoemulsions are conferred by the low surface tension of the system [21].

### 4.3.2. Skin Hydration and Oiliness, and pH Evaluation

Hydration of Skin—Activity Evaluation

As we can see in Figure 5A, at 30 min F38J (with vegetable extracts) show the greater hydration values while the commercial product the smallest hydration power. The vegetable extracts collaborate to this fact and the F38 show the medium hydration value power. No significant statistical difference was observed between formulations hydration power at all analized times. The suggested nanoemulsions have major hydration values when compared to industrial product.

Commercial nanoemulsion comprises wetting agents and emollients including vegetable oil, glycerin, biosacharide gum, cetyl palmitate, silicone and sorbitol. However, the presence of several moisturizers compounds, do not provide significant higher hydration in relation to babassu oil nanoemulsions.

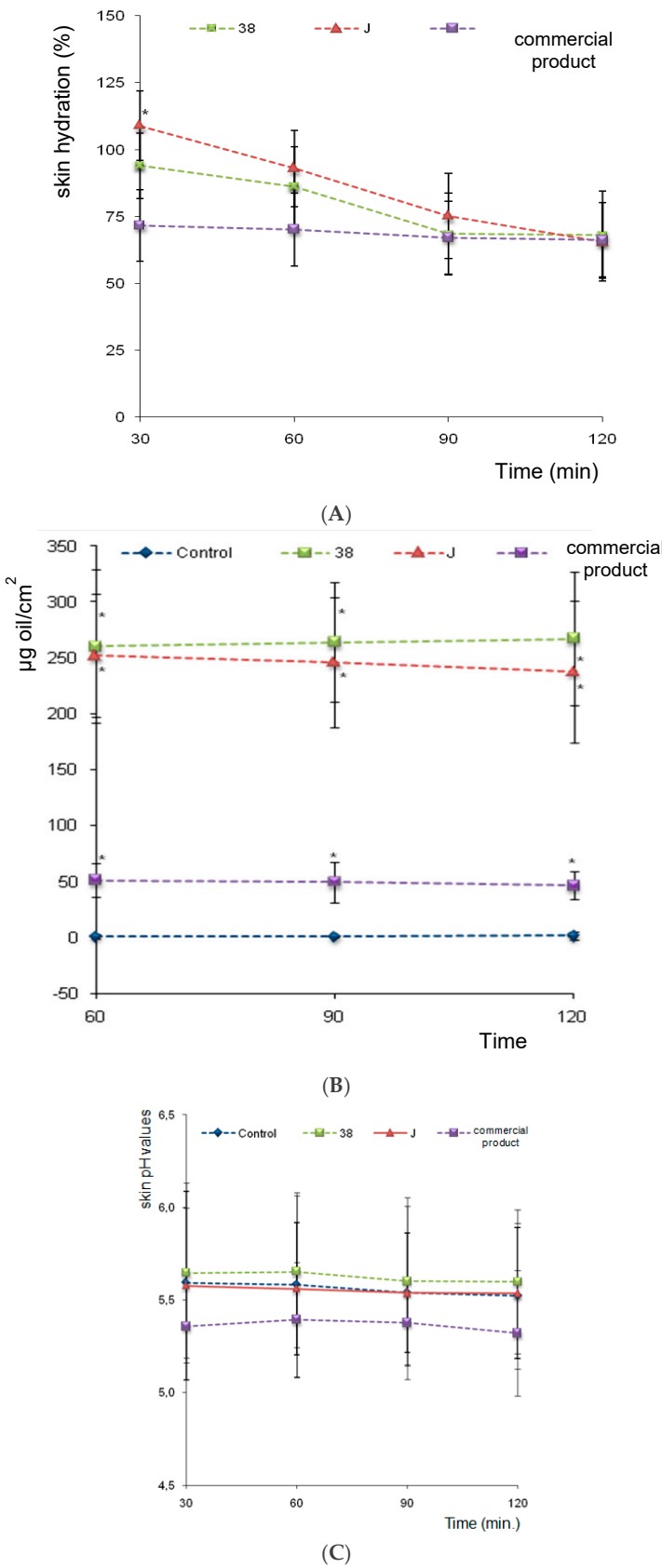

**Figure 5.** Hydration activity (**A**); skin oiliness (**B**); and skin pH evaluation (**C**) as a function of time. (*) no statistical differences between F38 and F38J. (*) Significance level compared to the control group (**C**) (*p* < 0.001).

Skin Oiliness Evaluation

The skin oiliness values for control regions did not change and were maintained around 0.0 to 1.0 mg/cm$^2$. Both F-38 and F-38J babassu oil nanoemulsions promoted an increase in the amount of skin oils, making statistically significant increases over the control area at 30, 60, 90, and 120 min. There was no statistical difference between F-38 and F-38J formulations, however, they diverge from the commercial nanoemulsions, which showed skin oiliness values about four times lower (Figure 5B).

Skin pH Evaluation

No statistically significant changes for nanoemulsions were observed in skin pH values, maintained at about 5.5, i.e., close to the normal pH value of the control area (4.2 to 5.9), indicating that the applied formulations were suitable for cosmetic use (Figure 5C).

## 5. Conclusions

Stable babassu oil nanoemulsion were obtained with surfactant pair fatty ester/PEG-54 castor oil with an HLB$_{required}$ value = 10.0 and having a droplet size of 45.97 $\pm$ 1.43 nm. The composition, order addition of components, and process parameters such as temperature, speed, and time of agitation, were critical in obtaining stable babassu oil nanoemulsions. There was no observed change in the size for both F-38 and F-38J nanoemulsions.

Although there are differences between the results of the HET-CAM and RBC tests, formulations F-38 and F-38J may be indicated for cosmetic use. In addition, both showed ear edema reduction capacity and skin moisturizing with no influence on skin pH value. Our results suggest the potential cosmetic use of babassu oil nanoemulsions.

**Acknowledgments:** This work had financial support from CAPES (Coordenação de Aperfeiçoamento de Pessoal de Nível Superior) and FAPESP (Fundação de Amparo a Pesquisa do Estado de São Paulo) under the protocol number (2009/07817-7; 2009/06152-1). We thank Lipo do Brasil, Croda and Oxiteno for supplying the raw material.

**Author Contributions:** V.C.G. was the student who carried out the laboratory work, analyzed the data and drafted the paper. O.F.S. contributed with the laboratory work; M.F. and M.M. contributed with the analysis of the data and the critical reading of the manuscript; P.A.R.-F. was the principal scientific supervisor of the study. He designed the study, supervised the laboratory work and contributed to critical reading of the manuscript.

**Conflicts of Interest:** The authors declare no conflict of interest.

**Ethics Approval and Consent to Participate:** All human experiments were implemented with approval of the Ethics Committee for Human Experiments (CEP/FCFRP n$^r$ 204/2011). For HET-CAM analysis it's not necessary if the eggs have less than 15 days embryonated.

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
