# Peer review of "In Vitro and In Vivo Evaluation of Nanoemulsion Containing Vegetable Extracts"

_cosmetics, doi:10.3390/cosmetics4030032_

Round 1

Reviewer 1 Report

The authors report the in vitro and in vivo evaluation of nanoemulsions containing vegetable extracts. The subject is certainly important for dermal applications of nanoemulsions. I think the manuscript in its present form is a very poor one and is probably publishable after major revision.

Line 22: Please replace “droplet size of 45.97±1.43nm” with “droplet size of 46±1nm”. In addition it should be mentioned if we speak about diameter or radius.

Lines 27-38: There are several errors in these lines. Please improve English language style.

Lines 40-55: The Introduction should be rewritten taking into consideration the following remarks: The purpose of the work and its importance in the field should be emphasized. The current state of the research field should be reviewed carefully and key publications cited.

Line 56: Replace “Material” by “Materials”

Line 60: More information should be given about the vegetable extracts. Are they oily or water soluble? The nature of the solvent and the extraction process should be also mentioned.

Methods: Authors should add new paragraphs to explain how particle size distribution, polydispersity index, pH, electrical conductivity and refractive index values were evaluated?

Lines 185-232: There are several errors in these lines. Please improve English language style.

Table 2: what is the meaning of using so many decimals?

The manuscript is a report of results. There is no discussion. Authors should improve their manuscript by COMPARING AND CONTRASTING THE NEW RESULTS WITH PREVIOUS ESTABLISHED LITERATURE.

Author Response

Thank you for the comments. We are reviewing the English text.

Reviewer 2 Report

It will be Interesting to control the activity of the effectiveness of the emulsions in vitro on keratinocytes culture to control their safeness and effectiveness

Author Response

(The authors gave the same response as above.)

Round 2

Reviewer 1 Report

Your manuscript has been considerably improved and can be now published in Cosmetics